# Integrated virtual screening and MD simulation study to discover potential inhibitors of mycobacterial electron transfer flavoprotein oxidoreductase

Kaleem Arshad[1,2]*, Jahanzab Salim[2], Muhammad Ali Talat[2], Asifa Ashraf[2], Nazia Kanwal[1]

1 Biological Sciences, Superior University, Lahore, Pakistan, 2 Khawaja Muhammad Safdar Medical College, Sialkot, Pakistan

* kaleemarshad630@gmail.com

## Abstract

Tuberculosis (TB) continues to be a major global health burden, with high incidence and mortality rates, compounded by the emergence and spread of drug-resistant strains. The limitations of current TB medications and the urgent need for new drugs targeting drug-resistant strains, particularly multidrug-resistant (MDR) and extensively drug-resistant (XDR) TB, underscore the pressing demand for innovative anti-TB drugs that can shorten treatment duration. This has led to a focus on targeting energy metabolism of *Mycobacterium tuberculosis* (Mtb) as a promising approach for drug discovery. This study focused on repurposing drugs against the crucial mycobacterial protein, electron transfer flavoprotein oxidoreductase (EtfD), integral to utilizing fatty acids and cholesterol as a carbon source during infection. The research adopted an integrative approach, starting with virtual screening of approved drugs from the ZINC20 database against EtfD, followed by molecular docking, and concluding with molecular dynamics (MD) simulations. Diacerein, levonadifloxacin, and gatifloxacin were identified as promising candidates for repurposing against TB based on their strong binding affinity, stability, and interactions with EtfD. ADMET analysis and anti-TB sensitivity predictions assessed their pharmacokinetic and therapeutic potential. Diacerein and levonadifloxacin, previously unexplored in anti-tuberculous therapy, along with gatifloxacin, known for its efficacy in drug-resistant TB, have broad-spectrum antimicrobial properties and favorable pharmacokinetic profiles, suggesting potential as alternatives to current TB treatments, especially against resistant strains. This study underscores the efficacy of computational drug repurposing, highlighting bacterial energy metabolism and lipid catabolism as fruitful targets. Further research is necessary to validate the clinical suitability and efficacy of diacerein, levonadifloxacin, and gatifloxacin, potentially enhancing the arsenal against global TB.

**Data Availability Statement:** All relevant data are within the manuscript and its Supporting information files.

**Funding:** The author(s) received no specific funding for this work.

**Competing interests:** The authors have declared that no competing interests exist.

## Introduction

Tuberculosis (TB), caused by *Mycobacterium tuberculosis* (Mtb), is a leading communicable disease and a significant contributor to global morbidity and mortality. In 2022, it was the second leading cause of death from a single infectious agent after COVID-19, with 7.5 million newly diagnosed cases and an estimated 10.6 million people developing it globally [1]. The disease resulted in approximately 1.30 million deaths, while 410,000 people developed multidrug-resistant or rifampicin-resistant TB (MDR/RR-TB) [1]. The high incidence and mortality rates of TB highlight the limitations of current medications, exacerbated by drug-resistant strains of Mtb that resist both first line (MDR) and second line (XDR) treatments [2]. Drug resistance in TB entails impeding prodrug activation, drug inactivation, extrusion, bacterial metabolic adaptation, target protein modifications, membrane pump alterations, altered drug interactions, and chromosome mutations [3, 4]. The pressing need for innovative anti-TB drugs is underscored by the shortcomings of current therapies such as side effects, expense, and high failure rates, compounded by the prolonged chemotherapy for TB, particularly MDR-TB, fostering drug resistance; thus, there's an urgent demand for new drugs targeting MDR and XDR strains, while shortening treatment duration for both drug-sensitive and drug-resistant TB [5, 6]. The rise and dissemination of drug resistant Mtb, coupled with the challenge of developing novel antimicrobials to counter resistance, have spurred scientists to explore new targets for drug discovery. Most current antimycobacterial drugs target the synthesis of nucleic acids, proteins, or folic acid, a focus that may have hindered the discovery of new therapeutics and contributed to drug resistance [7]. However, energy metabolism has garnered attention as a promising target for antibiotic drug development in mycobacteria, highlighted by the emergence of several drugs in clinical and preclinical stages focusing on bioenergetics, such as inhibitors of cytochrome $bc_1$:$aa_3$, NADH dehydrogenase, menaquinone synthesis, and ATP synthase [8].

Mtb has evolved to inhabit and interact with the human immune system, its primary host and reservoir, adapting its physiology to fulfill both cellular and pathogenic roles [9]. Mtb relies on host cholesterol and fatty acids as its primary carbon sources during infection, with a particular emphasis on cholesterol metabolism, especially in the chronic non-replicating phase, to prevent the accumulation of toxic long-chain fatty acids [10, 11]. Mtb adeptly utilizes host fatty acids and cholesterol in both intracellular and extracellular environments to generate energy, construct its lipid-rich cell wall, and produce virulence factors, facilitating its survival and pathogenesis [12]. Mtb efficiently utilizes lipids through β-oxidation pathways to generate acetyl-CoA and propionyl-CoA, which are then channeled into central metabolic pathways for energy production and biosynthesis, while mitigating carbon loss and toxicity risks [12, 13]. Mtb encodes numerous enzymes for distinct steps of beta-oxidation, including about 35 acyl-CoA dehydrogenases (ACADs) [11]. Notably, all these ACADs utilize the electron transfer flavoprotein-oxidoreductase system as their electron acceptor. The electron transfer flavoprotein-oxidoreductase system is an evolutionarily conserved component of the electron transport chain, essential for energy production [14]. It functions by receiving electrons from FAD-containing acyl-CoA dehydrogenases and shuttling them to menaquinone, a liposoluble electron carrier [15]. This process is pivotal for the efficient transfer of electrons within the bacterial respiratory chain, contributing significantly to Mtb's energy metabolism. Within Mtb, a critical enzyme complex oversees this transfer of electrons from ACADs to menaquinone, comprising an electron transfer protein with two subunits, FixA (Rv3029c) and FixB (Rv3028c), along with a membrane-bound electron transfer flavoprotein-oxidoreductase, EtfD (Rv0338c) [11]. Deletion of the gene Rv0338c, encoding EtfD, disrupts β-oxidation at the step catalyzed by ACADs, rendering mutants deficient in Rv0338c unable to grow on fatty acids

and cholesterol [11]. Additionally, these mutants lacking Rv0338c are susceptible to the bactericidal effects of long-chain fatty acids, which impair growth and survival in mice, aligning with the established fact that long-chain fatty acids are bactericidal due to their inhibition of crucial bacterial enzymes such as FabI [11, 16]. Moreover, the accumulation of reduced flavin adenine dinucleotide (FAD) induces reductive stress, impairing metabolism, causing protein aggregation, generating reactive oxygen species (ROS), and leading to cell death in mycobacterial cells [17]. Furthermore, inhibiting fatty acid catabolism triggers macrophage activation by inducing mitochondrial reactive oxygen species, enhancing macrophage NADPH oxidase and xenophagy activity, and consequently bolstering antimicrobial activity against Mtb infection [18]. The gene Rv0338c has been deemed indispensable through Tn saturation mutagenesis and is presumed to encode a putative hetero-disulfide reductase containing iron-sulfur, likely participating in energy production and conversion [19, 20]. Further confirmation of Rv0338c as an anti-Mtb drug target was achieved with the discovery of mutations in response to DBPI compounds, conferring resistance and occurring near iron-sulfur domains [19]. Highlighting the criticality of targeting bioenergetics in Mtb, this discussion emphasizes EtfD's pivotal role in lipid metabolism, where its disruption not only impairs Mtb's growth on fatty acids and cholesterol, enhancing susceptibility to long-chain fatty acids, but also augments macrophage antimicrobial activity, thereby presenting a compelling therapeutic target.

Mitigating toxicity in drug development necessitates antibacterials that selectively inhibit mycobacterial energy metabolism to avoid target-based toxicity from shared bacterial-eukaryotic pathways [6]. Drug repurposing emerges as a vital strategy in combating TB by leveraging drugs with established safety profiles and shorter regulatory paths, particularly through targeting novel pathways like bioenergetics with promising results [21]. This study aims to repurpose drugs approved in major jurisdictions around the world to target EtfD in Mtb by employing structure-based drug design, a computer aided methodology. Through this approach, we seek to identify promising therapeutic candidates that can effectively inhibit EtfD, offering a novel and potent strategy against drug-resistant TB.

## Material and methods

This study utilized various computational tools: Discovery Studio Visualizer 2020, and PyMOL 3.0 for protein preparation and interaction visualization, COACH online metaserver for ligand binding site prediction, PyRx 0.8 for virtual screening, AutoDock4 (AD4) and AMBER for molecular docking and dynamics simulations, SwissADME and pkCSM for ADMET analysis, and mycoCSM for predicting antituberculosis sensitivity (Fig 1) [22–29].

### Structural evaluation of EtfD

The three-dimensional structure of the EtfD (ID: O33268) was retrieved from the AlphaFold Protein Structure Database (AlphaFold DB) [30]. The predicted structure was evaluated for reliability using predicted local distance difference test (pLDDT), and predicted aligned error (PAE) scores, MolProbity, flDPnn, ProQ, ProSA, and InterPro tools [31–37]. This comprehensive evaluation ensures the structure's suitability for further studies.

### Binding site prediction and characterization

The COACH metaserver was employed to predict the ligand binding site of EtfD, followed by a detailed examination of its stability and interactions with ligand.

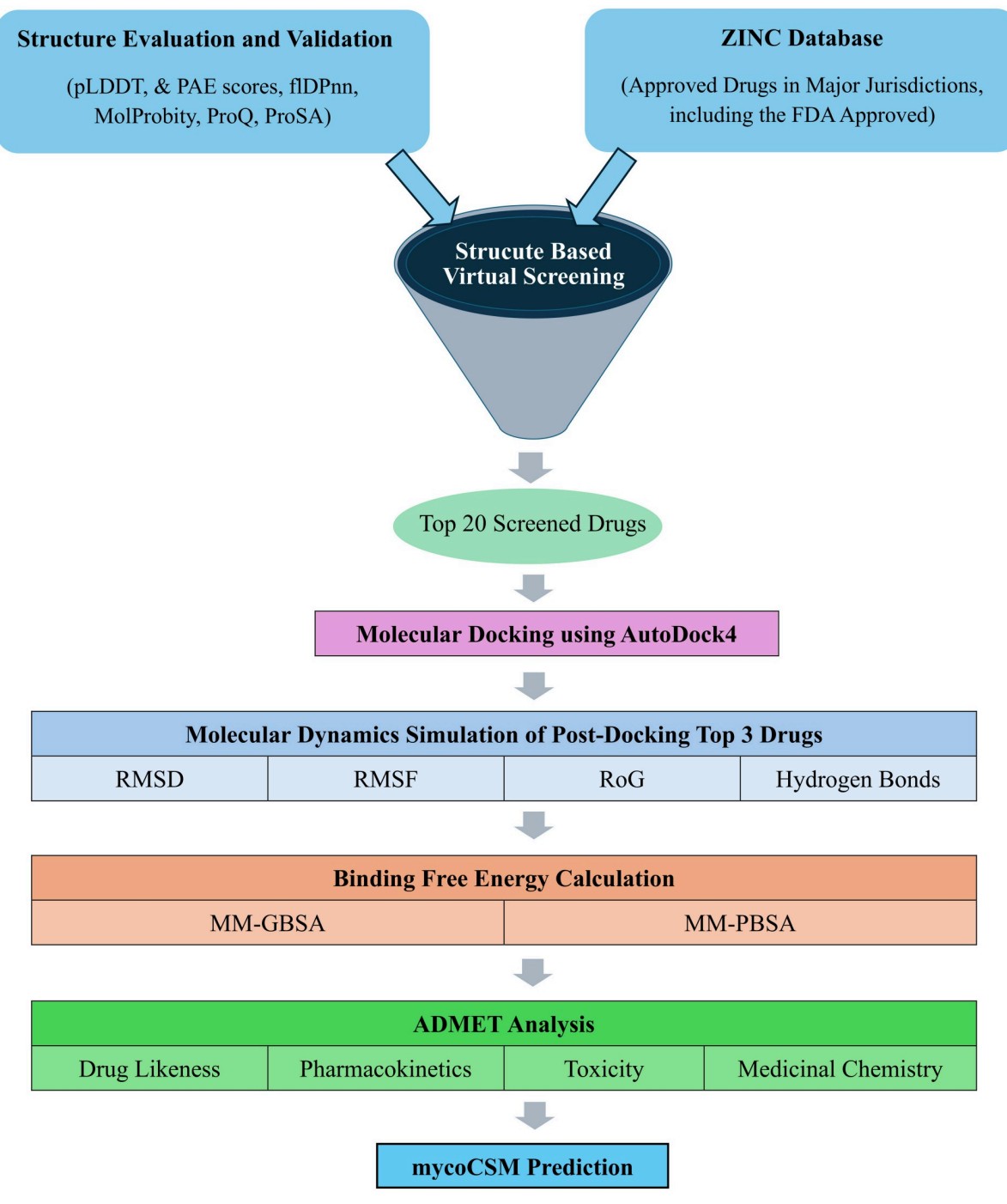

**Fig 1. Methodology flowchart.** pLDDT, predicted local distance difference test; PAE, predicted aligned error; flDPnn, putative function- and linker-based Disorder Prediction using deep neural network.

## Structure based virtual screening

**Protein preparation.** Discovery Studio Visualizer 2020 was utilized to prepare the EtfD. Using the "Define & Edit binding site" option, the EtfD and predicted ligand (iron sulfur cluster) were selected to define the binding site as a sphere keeping the ligand as the centroid. The

XYZ coordinates of the binding site sphere centroid were noted down for further use. The ligand was removed, polar hydrogens were added to the EtfD, and the protein structure was stored in PDB format.

**Ligand preparation.** About 3447 drug molecules in SDF format were downloaded from the ZINC20 database's "world" filter, which contains approved drugs in major jurisdictions, including the FDA approved [38]. The ligands were imported into the PyRx dashboard, where energy minimization and conversion into pdbqt format were executed using OpenBabel.

**Virtual screening.** The prepared EtfD protein structure was loaded to the dashboard and converted into autodock ligands in pdbqt format for input by PyRx for virtual screening. All 3447 ligands, along with the macromolecule, were defined within the Vina wizard. The auto grid box was configured using grid box dimensions of X = 18.6943 Å; Y = 19.3218 Å; Z = 21.9263 Å and previously obtained XYZ centroid coordinates (X = 4.2938; Y = 5.8156; Z = -6.6638). The screening procedure involved docking all ligands against the EtfD protein using the autodock vina wizard. The top 20 molecules were selected based on binding affinity for subsequent molecular docking.

## Molecular docking by AD4

Standard AutoDock protocols were applied for both receptor and ligand preparation, facilitated by AutoDockTools 1.5.7 software [39]. The grid center was set to specific XYZ centroid coordinates (X = 4.2938, Y = 5.8156, and Z = -6.6638), determined in prior virtual screening. The grid dimensions were set to 60 × 60 × 60 with a spacing of 0.375 Å, resulting in an AD4 grid size of 22.5 Å × 22.5 Å × 22.5 Å. Grid parameters were generated using Autogrid4, and the docking parameter file was generated using AutoDockTools. Briefly, the EtfD was configured for rigid docking, utilizing the genetic algorithm, with all other docking parameters maintained at their default values. Notably, the number of genetic algorithm (GA) runs, and the maximum number of energy evaluations (eval) were specifically set to 250 and 25,000,000 (which correspond to "long" option) respectively for molecular docking calculations. The AD4 program was then executed to obtain docking results, revealing the corresponding binding energies. The three drugs selected for subsequent molecular dynamic (MD) simulation based on their highest binding affinity were diacerein (ZINC ID: ZINC000003812842), levonadifloxacin (ZINC ID: ZINC000000603195), and gatifloxacin (ZINC ID: ZINC000003607120).

## MD simulation

The top 3 docked drugs were subjected to MD simulations for investigating intermolecular affinity in dynamics on a time scale of 100 ns, utilizing AMBER22. The FF19SB force field was employed to prepare parameters of the EtfD while AMBER General Force Field 2 (GAFF2) was used for drugs processing [40, 41]. The Antechamber tool of AmberTools23 was used to generate parameters missing in GAFF2 to process drugs [42]. The EtfD was submitted to the H++ server (http://biophysics.cs.vt.edu/H++) to predict its protonation state, resulting in the addition of six Na+ counterions based on the predicted charge (-6 at pH 7.4), and the complexes were then placed into a 10 Å truncated octahedron box of OPC water molecules [43, 44]. The equilibration step for MD systems utilized pmemd from AMBER22 for energy minimization and pmemd.cuda for relaxation processes and final MD simulation run. The water and ions were energy minimized for 1000 steps while everything else in the systems was restrained. Then the systems were heated from 100K to 298 K over 1000 ps using Langevin dynamics. Post-heating, the systems were relaxed under constant pressure for 1000 ps to facilitate density and volume equilibration, followed by a further round of energy minimization for 1000 steps. This was followed by relaxation for 2000 ps, wherein positional restraints on the

protein backbone and ligands were progressively reduced leading to their complete removal in the last 1000 ps of the relaxation, and production run of 100 ns. Post-simulation analyses, including Root Mean Square Deviation (RMSD), Root Mean Square Fluctuation (RMSF), Radius of Gyration (ROG), and hydrogen bond analysis, were conducted using the AMBER CPPTRAJ module to assess the stability of complexes and visualize structural deviations over time [45–47].

## Binding free energy estimation by MM-G/PBSA

The MMPBSA.py package from AMBER22 was utilized to estimate the binding free energies of the systems [48]. The primary objective of this analysis was to determine the difference in free energy between two states of the complex, namely solvated and gas phase, using the following equation.

$$\Delta G_{bind,\ solv} = \Delta G_{bind,vacuum} + \Delta G_{solv,\ complex} - \left( \Delta G_{solv,\ ligand} + \Delta G_{solv,\ receptor} \right)$$

From the complete simulation trajectories, 100 frames were selected as input for both MM/PBSA and MM/GBSA calculations. The selection of these 100 frames was facilitated using an input parameter file of AMBER22 MM-GB/PBSA, which enabled the consideration of frames picked at equal time intervals from the simulation trajectories.

## ADMET analysis

Physicochemical properties, medicinal chemistry, druglikeness, and ADMET analysis of the top drugs were performed through online servers i.e. pkCSM and SwissADME [26, 27].

## Anti-TB sensitivity prediction

We utilized the mycoCSM online server for predicting the anti-TB activity of the top 3 drugs [28]. The compounds' SMILES format was uploaded, and the analysis report was downloaded in CSV format.

# Results

## Structural evaluation of EtfD

AlphaFold provides pLDDT and PAE metrics to assess predicted model's accuracy. The pLDDT score, ranging from 0 to 100, estimates the agreement between predicted and experimental protein structures, effectively assessing local model quality, with lower scores typically correlating with a higher likelihood of intrinsic disorder [31]. The predicted model of the EtfD has an average pLDDT value of 82.56, indicating overall high confidence, with specific regions exceeding 90. However, regions 342 to 357 and 745 to 882 display low values below 50, suggesting likely disorder (Fig 2). PAE indicates the predicted error between relative positions of residue pairs in protein structures, with low values suggesting well-defined relationships and high values indicating unreliable positions [49]. The PAE 2D heatmap shows low confidence in the positions of residues in regions 342–357 and 745–882 relative to the rest of residues, which also have low pLDDT values, lack secondary structure, and exhibit coiled, ribbon-like appearance, predicting disorder (S1 Fig). The structure similarity clustering in the AlphaFold Protein Structure Database, using MMseq2 and Foldseek, identified a cluster related to EtfD, comprising proteins with iron-sulfur binding domains and oxidoreductase activity across diverse bacterial species, some also with cysteine-rich domains [50, 51]. This highlights structural homogeneity among EtfD-related proteins based on their shared iron-sulfur clusters and oxidoreductase functions.

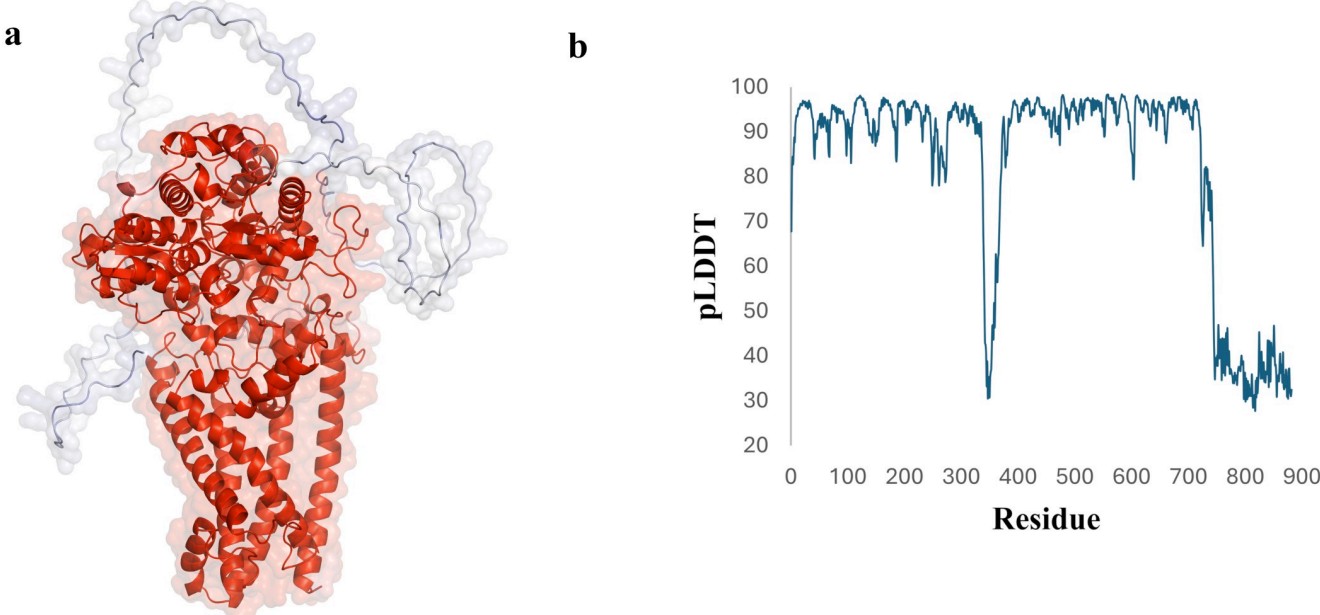

**Fig 2. Evaluation of EtfD's predicted structure.** (a) EtfD's structure colored by pLDDT scores, with red indicating high confidence and blue indicating low confidence. The cartoon representation is overlaid with a semi-transparent surface. (b) pLDDT score plot.

The EtfD protein, consisting of 882 amino acids, is likely involved in energy production and conversion, as indicated by InterPro (https://www.ebi.ac.uk/interpro/) [37]. It features an N-terminal domain (residues 5–231) with six transmembrane segments, followed by a [4Fe-4S] ferredoxin-type iron-sulfur binding domain (residues 286–412, IPR017812). This is succeeded by two cysteine-rich domains (residues 477–568 and 607–693, IPR004017). The C-terminal domain is characterized by a proline—alanine-rich, repetitive structure of low complexity and is predicted to be intrinsically disordered.

Intrinsically disordered regions (IDRs) lack a stable three-dimensional structure due to low hydrophobicity and high net charge but can adopt various configurations upon ligand binding based on amino acid composition and charge patterning, allowing them to efficiently interact with multiple targets under physiological conditions [52–54]. The flDPnn, a disorder prediction tool utilizing deep neural networks, accurately predicts disorder and fully disordered proteins, while also generating putative functions for predicted IDRs [34]. The flDPnn server identified same regions as IDRs that corresponded with low pLDDT and high PAE scores from AlphaFold. However, it also predicted these regions to have putative functions in binding with various biomolecules (S1 Fig).

MolProbity is a tool used to validate and analyze biomolecular 3D structures by assessing factors like geometry, steric clashes, and overall structural validity [33]. MolProbity analysis of EtfD revealed 93% of residues in the favored region and additional 4.6% in the allowed region of the Ramachandran plot, with 21 outliers observed. Notably, 20 out of 21 outliers corresponded to residues within predicted disordered regions (S2 Fig). The Ramachandran distribution Z-score for EtfD was -1.55 ± 0.26, indicating a reasonable agreement with typical protein structures and suggesting that the predicted dihedral angles are within acceptable ranges. The ProSA results show that the Z-score for the predicted EtfD model is -10.15, which falls within the range of native conformations (S3 Fig). The overall residue energies are predominantly negative, except for some peaks observed in the N-terminal transmembrane

regions and the C-terminal disordered region (S4 Fig). Additionally, the ProQ LG score of 7.147 suggests that the predicted EtfD model closely resembles native protein structures.

The overall confidence in EtfD's predicted structure is high, except for two disordered regions with the lowest pLDDT scores, which also exhibit uncertain positions relative to the protein, as indicated by high PAE values. However, the rest of the structure shows high reliability, with many regions having pLDDT values over 90, comparable to experimental structures. This accuracy supports various applications, especially for precise tasks like characterizing binding sites. The connecting loops also have favorable pLDDT values (70–90), indicating good backbone prediction. This assessment confirms the model's reliability, making it suitable for further research, especially in drug design against this protein.

## Binding site prediction and characterization

Identifying a protein's ligand binding site is essential for understanding its function and designing therapeutic compounds to modulate its activity [55]. The EtfD model was submitted to the COACH server, employing a metaserver approach to predict potential ligand binding sites and propose probable ligands interacting with the protein of interest. COACH ranks the predicted ligand binding sites based on the C-score, a confidence score ranging from 0 to 1, where a higher score indicates a more reliable prediction. COACH detected several ligand binding sites, and the site ranked first, with the highest confidence score of 0.20, was chosen for subsequent analysis (Fig 3a and 3b). The COACH server analysis revealed that the key amino acid residues constituting the active site of EtfD include CYS 295, THR 296, GLU 297, CYS 298, GLY 299, CYS 301, LYS 317, CYS 402, PRO 403, and ILE 406 (Fig 3c). It is crucial to note that the confidence score for this binding site is relatively low. However, it is noteworthy that individual algorithms, such as TM-Site (Score: 0.35), S-Site (Score: 0.30), and ConCavity (Score: 0.55), gave relatively high scores for almost the same amino acids predicted to be binding site residues. However, considering the important findings discussed in previous sections, it is reasonable to expect that the predicted residues constitute a binding site for the iron-sulfur cluster, and these predictions align with our expectations.

The predicted ligand, Iron Sulfur Cluster (SF4), directly interacts with residues THR 296, CYS 298, GLY 299, CYS 301, CYS 402, and HIS 408. Although not predicted as a binding site residue, HIS 408 plays a crucial role, engaging in two pi-sulfur interactions and a hydrogen bond with two sulfur atoms of SF4. Additional significant interactions of SF4 include hydrogen bonds with CYS 301, GLY 299, CYS 298, and two with THR 296. SF4 forms coordinate bonds with CYS 295, CYS 298, and CYS 301, with the fourth iron atom participating in a metal acceptor interaction with CYS 402 (Fig 3d and S5 Fig).

Examining the intricate interactions between binding site and its surroundings is crucial for understanding structural stability and functional significance. Binding site residues engage in diverse interactions with surrounding residues, encompassing hydrogen bonds, electrostatic interactions, alkyl interactions, and salt bridges. Notably, Glutamate 297, Glycine 299, Lysine 317, and Cysteine 402 are involved in multiple hydrogen bonds with surrounding residues (S5 Fig).

The identification of the predicted binding site in EtfD, along with its probable ligand, an iron sulfur cluster, provides compelling evidence that supports our initial assumptions about the protein's involvement in metabolism. Targeting this binding site can be a promising strategy for drug design against Mycobacterium tuberculosis (Mtb), underlining the potential significance of this structural insight in advancing therapeutic interventions.

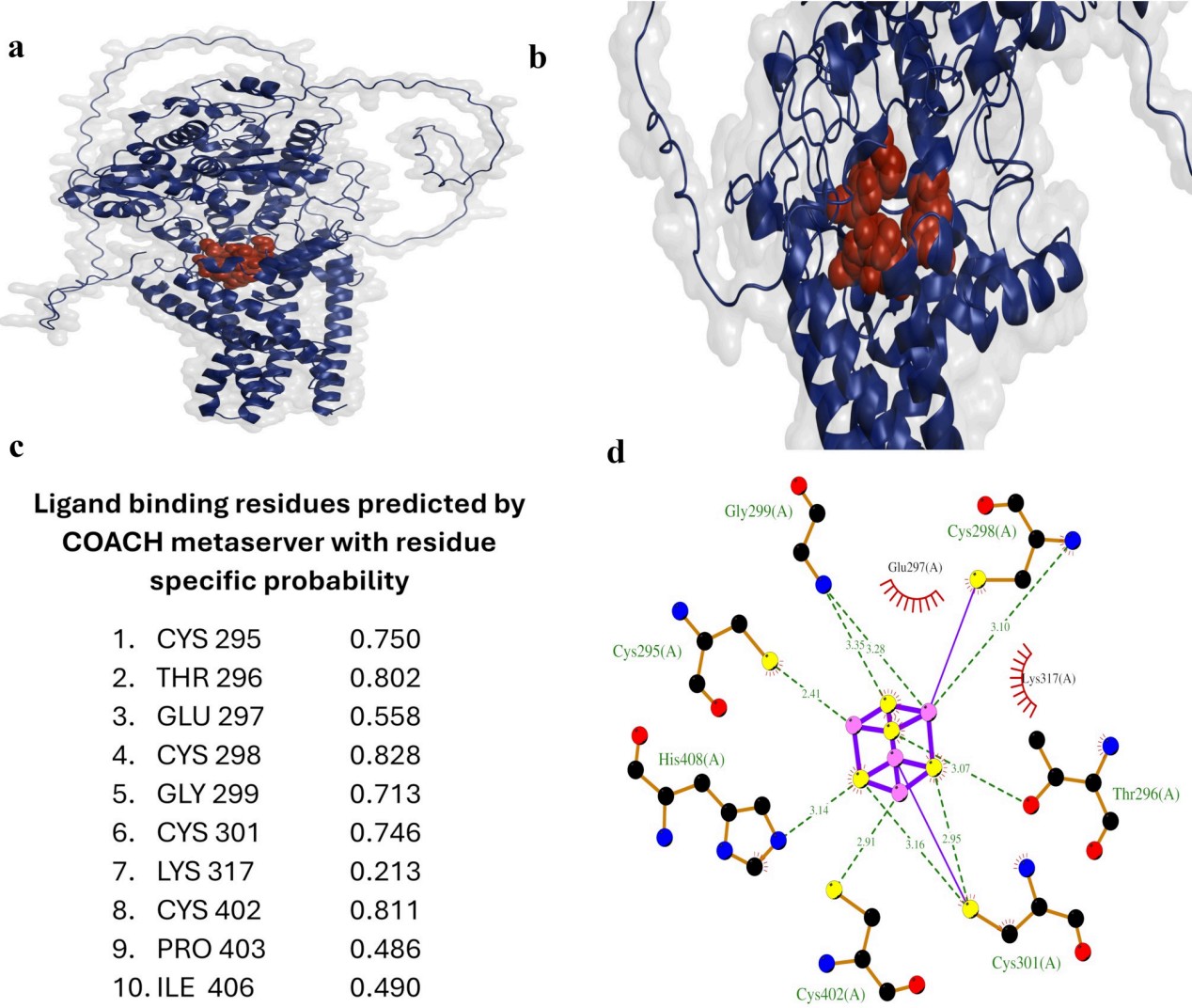

**Fig 3. EtfD's ligand binding site analysis.** (a) The overall protein structure is shown in blue as a cartoon, with the surface displayed. Binding site residues are highlighted in red and shown as balls. (b) Close-up view of the binding site. (c) Ligand binding site residues with their corresponding residue specific probabilities, as predicted by the COACH metaserver. (d) Predicted interactions between the ligand SF4 and binding site residues. Hydrogen bonds are shown in olive green, iron-sulfur coordinate bonds in purple, carbon in black, nitrogen in blue, oxygen in red, sulfur in yellow, and iron in pink. Hydrophobic interactions are not shown to avoid obscuring details, but the names of the residues involved in hydrophobic interactions are labeled in black.

### Virtual screening

A database of 3,447 approved drug molecules was screened against EtfD using PyRx 0.8. From Vina docking results, the top 20 compounds with binding affinities greater than -8.7 kcal/mol were prioritized for repurposing (S1 Table).

### Molecular docking

The top 20 molecules were re-docked against EtfD using AD4, with binding energies ranging from -7.31 to -10.47 kcal/mol (Table 1). The top hits for further MD simulation studies are ZINC000003812842 (diacerein), ZINC000000603195 (levonadifloxacin), and

**Table 1. Binding affinity scores for top 20 drug molecules (AutoDock4).**

| No. | ZINC ID/ drug name | AutoDock4 docking score (kcal/mol) |
|---|---|---|
| 1 | ZINC000003812842/ diacerein | -10.47 |
| 2 | ZINC000000603195/ levonadifloxacin | -10.04 |
| 3 | ZINC000003607120/ gatifloxacin | -10.02 |
| 4 | ZINC000003794622/ nadifloxacin | -9.94 |
| 5 | ZINC000000001894/ pefloxacin | -9.80 |
| 6 | ZINC000003873157/ lomefloxacin | -9.49 |
| 7 | ZINC000003919580/ formestane | -9.31 |
| 8 | ZINC000003875998/ isopregnenone | -9.21 |
| 9 | ZINC000000020220/ ciprofloxacin | -9.19 |
| 10 | ZINC000000000917/ amifloxacin | -9.01 |
| 11 | ZINC000000601275/ talniflumate | -8.91 |
| 12 | ZINC000004081771/ testolactone | -8.66 |
| 13 | ZINC000003812989/ nalbuphine | -8.36 |
| 14 | ZINC000000538285/ repirinast | -8.29 |
| 15 | ZINC000013509425/ estrone | -8.15 |
| 16 | ZINC000000119434/ strychnine | -7.82 |
| 17 | ZINC000003812988/ butorphanol | -7.79 |
| 18 | ZINC000003812889/ tibolone | -7.67 |
| 19 | ZINC000008143788/ artemisinin | -7.60 |
| 20 | ZINC000003779726/ pazufloxacin | -7.31 |

ZINC000003607120 (gatifloxacin), with binding energies of -10.47, -10.04, and -10.02 kcal/mol, respectively. Diacerein forms hydrogen bonds with ARG133, ARG146, and ARG300, and a carbon hydrogen bond with ALA155. It also exhibits hydrophobic interactions with HIS235 and CYS298, and electrostatic interactions with ARG146. Levonadifloxacin interacts with EtfD through hydrogen bonds involving ARG133, ARG146, ARG300, and GLY154, and a carbon hydrogen bond with ALA155. Hydrophobic interactions, including pi-alkyl and alkyl interactions, occur with HIS235, CYS298, and HIS74, enhancing stability. Gatifloxacin predominantly relies on hydrophobic interactions, involving key amino acids PHE129, ILE132, ARG133, PHE147, ALA155, VAL158, and LYS232. PHE129 forms four significant hydrophobic interactions. Additionally, ARG133 and ARG300 contribute to hydrogen bonding, along with ALA155. HIS74 and HIS235 facilitate halogen (fluorine) interactions, with HIS74 also engaging in a hydrogen bond. ARG146 stabilizes the interaction through electrostatic interactions (S6 Fig). Fig 4 visually illustrates the docking interactions between EtfD and diacerein, levonadifloxacin, and gatifloxacin.

## MD simulation analysis

MD is a computational technique modeling the dynamic behavior of molecular systems over time, treating all elements as flexible, and is typically used to further investigate the highest-ranked complex for detailed exploration [56]. Two widely used metrics for assessing structural fluctuations of macromolecules are Root-Mean-Square Deviation (RMSD) and Root-Mean-Square Fluctuations (RMSF). RMSD quantifies the average displacement of atoms from a reference structure, aiding in the analysis of time-dependent structural motions, often indicating stability or divergence, which may suggest simulation non-equilibration [45]. Whereas RMSF measures the displacement of specific atoms or groups from the reference structure, indicating

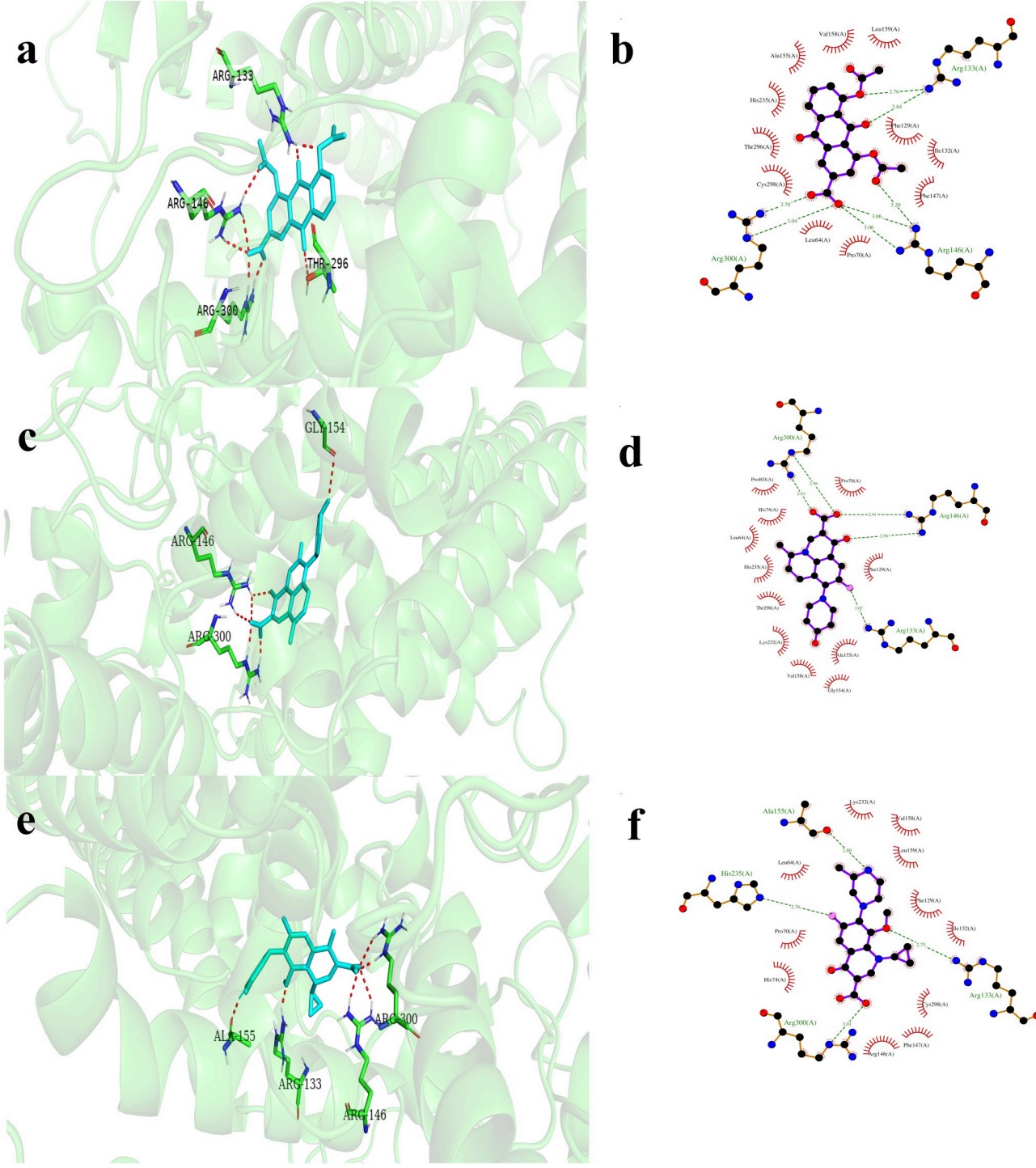

**Fig 4. Molecular models showing binding interactions between EtfD and diacerein, levonadifloxacin, and gatifloxacin in 3D and 2D.** (a and b) EtfD-diacerein, (c and d) EtfD-levonadifloxacin, (e and f) EtfD-gatifloxacin. EtfD hydrogen-bonding residues are shown as sticks (carbon: green, nitrogen: blue, oxygen: red), with hydrogen bonds represented by red dashed lines. 2D interaction diagrams follow the same color scheme as described in Fig 3d. Common amino acids involved in interactions with EtfD across diacerein, levonadifloxacin, and gatifloxacin are ARG133, ARG146, ALA155, HIS235, and CYS298.

protein flexibility during a simulation [45]. RMSD and RMSF gauge protein mobility via rigid body alignment to a reference, but their sensitivity to fluctuating subsets can inflate values, potentially misrepresenting overall dynamics [45]. The Radius of Gyration (RoG) is pivotal in assessing amino acid residue packing for protein stability, with conformational changes upon ligand binding making its calculation crucial for predicting macromolecular structural activity and stability [46, 47]. Hydrogen bonds and hydrophobic interactions are critical for stabilizing macromolecules, influencing binding affinity, drug efficacy, and providing essential insights for drug design and behavior prediction, with their paramount role in stabilizing the protein-ligand complex significantly impacting drug-target binding (Fig 5 and S7 Fig) [47].

**Diacerein.** The RMSD of the EtfD backbone complexed with diacerein gradually increased over the first 25 ns, followed by fluctuations until 45 ns, with a peak at 43 ns. It then fluctuated until 65 ns while gradually converging toward the average RMSD. Beyond 65 ns, the

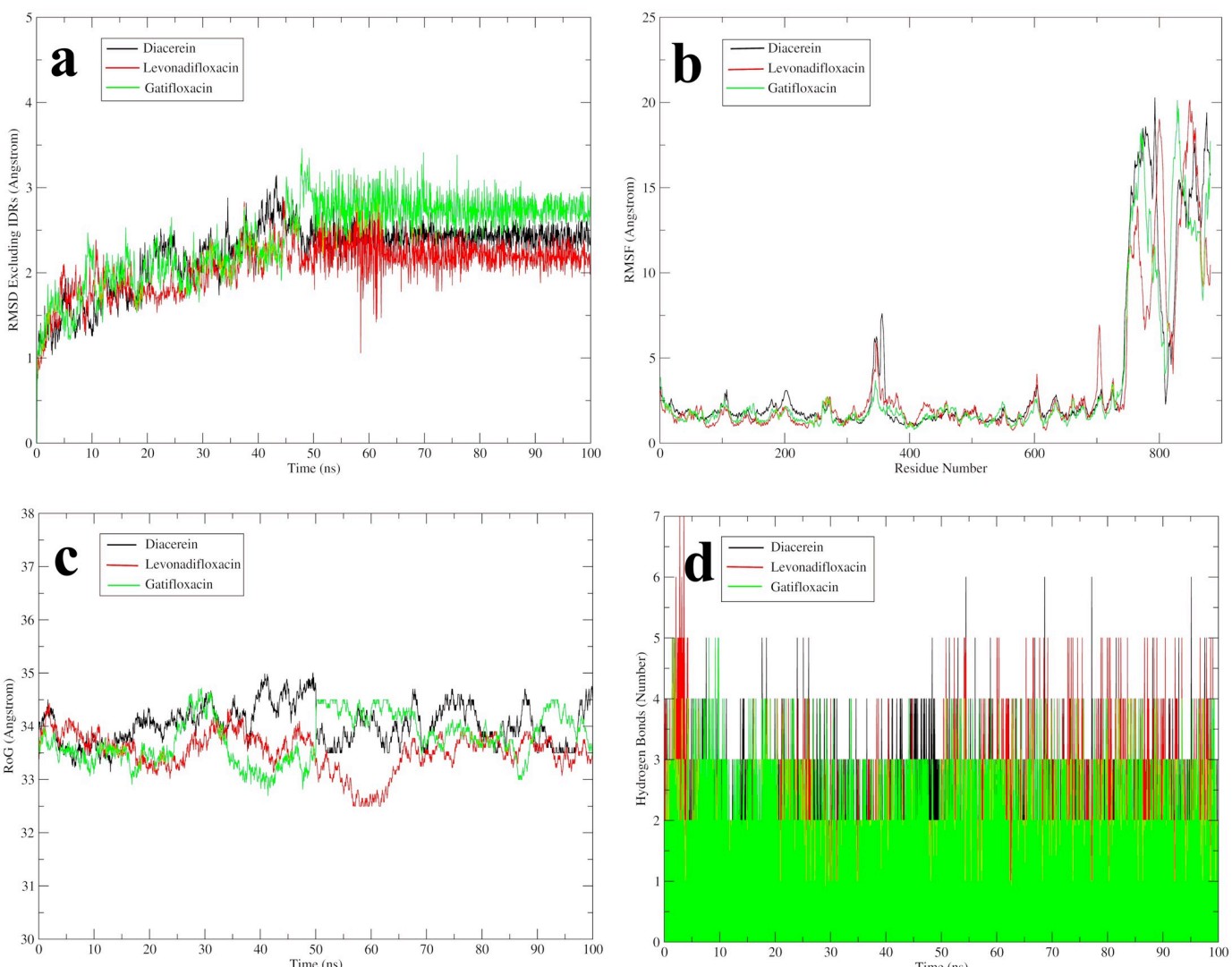

**Fig 5. Investigating the stability of EtfD-drug complexes through MD simulations using various statistical parameters.** (a) RMSD. (b) RMSF. (c) RoG. (d) hydrogen bonding. RMSD, root mean square deviation; RMSF, root mean square fluctuation; RoG, radius of gyration.

RMSD stabilized with only slight oscillations, indicating a steady conformation. The average RMSD of the backbone of EtfD, excluding IDRs, was calculated as 2.2315 ± 0.3845. Conversely, the full protein backbone RMSD displayed significant fluctuations, largely attributed to the IDRs, with a mean value of 6.0513 ± 3.928. Comparison of the two RMSD values indicates overall stability of the protein structure throughout the simulation, except for the IDRs which exhibited the highest fluctuations. Additionally, RMSF analysis confirmed this observation, with residues excluding IDRs displaying RMSF values of 1.8149 ± 0.5736, while IDRs exhibited considerably higher RMSF values (12.6199 ± 4.6786). The Radius of Gyration (RoG) value for the bound complex was 34.0314 ± 0.3725, suggesting stability throughout the simulation, with minor fluctuations throughout simulation within acceptable limits. The presence of numerous hydrogen bonds between diacerein and EtfD throughout the simulation, involving residues HID74, ARG133, ARG146, THR296, and ARG300, underscores the high stability of the complex, with sustained interactions observed for ARG133 and ARG300.

**Levonadifloxacin.** The RMSD of the EtfD backbone, excluding IDRs, exhibited a gradual increase up to 20 ns, followed by a stable phase until 30 ns. Between 30 and 60 ns, significant fluctuations were observed, transitioning to moderate fluctuations until 80 ns, after which the RMSD converged. The average RMSD of the EtfD backbone, excluding IDRs, was 2.0655 ± 0.3063 Å. Conversely, RMSD of the full protein's backbone exhibited notable fluctuations primarily due to IDRs, with a recorded value of 5.7226 ± 3.7903 Å, highlighting the overall stability of the protein structure throughout the simulation, except for IDRs with high fluctuations. The RMSF analysis further validated these observations, showing residues excluding IDRs with an RMSF value of 1.7463 ± 0.6670 Å, while residues within IDRs displayed notably higher RMSF values of 11.0471 ± 4.6931 Å. Additionally, the Radius of Gyration (RoG) value for the bound complex was determined to be 33.5237 ± 0.3619 Å, indicating stability throughout the simulation duration. LEU64, TRP67, PRO70, ARG133, ARG146, HIE233, and ARG300 amino acids formed hydrogen bonds with levonadifloxacin during the simulation. Notably, HIE233, ARG146, PRO70, ARG133, and ARG300 maintained consistent interactions with levonadifloxacin throughout, with HIE233 being the most prominent, followed by ARG146, PRO70, ARG133, and ARG300.

**Gatifloxacin.** The RMSD of the EtfD backbone, excluding IDRs, gradually increased in the first 10 ns and remained stable with minor fluctuations until 45 ns. This was followed by a notable rise and extreme fluctuations up to 65 ns, after which the structure continued fluctuating until 90 ns, before stabilizing in the final 10 ns. The average RMSD of the EtfD backbone with gatifloxacin was 2.1685 ± 0.5704 Å, indicating stability, while the full protein backbone exhibited high fluctuations primarily due to IDRs (4.7926 ± 3.3495 Å). Comparative analysis showed overall protein structure stability except for IDRs, supported by RMSF values (residues excluding IDRs: 1.8163 ± 0.4962 Å, IDRs: 11.2945 ± 4.2492 Å). RoG value (33.4583 ± 0.3799 Å) indicated stability in the bound complex throughout the simulation. Notably, HID74, ARG133, ARG146, HID151, ASN152, ALA155, TRP156, HID233, and ARG300 residues formed significant hydrogen bonds with gatifloxacin, highlighting the high stability of this complex. Particularly, ARG300 exhibited multiple hydrogen bonds with the drug, followed by HID74 and ARG146.

## Binding free energy estimation

To validate the drugs' affinity to EtfD, MM-GB/PBSA post-simulation processing was conducted to obtain different free energies of the complexes (Table 2).

**Table 2. MM-G/PBSA net binding energy of the molecules presented for each energy component.**

| Compound | ΔG binding | ΔG electrostatic | ΔG binding vdW | ΔG binding gas phase | ΔG polar solvation | ΔG non-polar solvation | ΔG solvation |
|---|---|---|---|---|---|---|---|
| MM-GBSA | | | | | | | |
| Diacerein | -47.6459 | -51.4789 | -42.0295 | -93.5084 | 51.9485 | -6.0861 | 45.8625 |
| Levonadifloxacin | -39.0626 | -32.9499 | -43.8081 | -76.7580 | 42.4935 | -4.7981 | 37.6954 |
| Gatifloxacin | -41.0665 | -34.0843 | -37.2723 | -71.3566 | 36.2007 | -5.9106 | 30.2901 |
| MM-PBSA | | | | | | | |
| Diacerein | -59.1380 | -49.9564 | -42.4629 | -92.4193 | 66.8367 | -33.5554 | 33.2813 |
| Levonadifloxacin | -43.0536 | -31.5991 | -42.0791 | -73.6782 | 61.2473 | -30.6227 | 30.6246 |
| Gatifloxacin | -59.3589 | -34.2496 | -37.1834 | -71.4330 | 44.0462 | -31.9721 | 12.0741 |

## ADMET analysis

Table 3 provides a comprehensive overview of the pharmacokinetics of the screened compounds, encompassing druglikeness, medicinal chemistry, and various toxicity analyses. The ADMET analysis of diacerein unveiled quinone_A (PAINS) and phenol_ester (Brenk) alerts, suggesting assay interference and chemical instability, while levonadifloxacin and gatifloxacin exhibited no alerts or issues While effective at identifying many aggregators and assay artifacts, the PAINS rule's broad application, absence of mechanism exploration, and incomplete validation contribute to its low precision and limited scope for detection [57]. Furthermore, the presence of a Brenk violation in diacerein due to the phenol ester group is mitigated by its metabolic conversion to rhein, which lacks such problematic substructures, thus these alerts can be safely ignored.

## Anti-TB sensitivity prediction

Certainly: Traditional drug testing against Mtb is challenging due to slow bacilli evolution, but machine learning tools like mycoCSM predict compound sensitivity preemptively (Table 4).

## Discussion

The emergence and persistence of drug-resistant strains of Mtb highlights the ongoing need for innovative approaches in TB research. Various strategies to discover new anti-tubercular agents include repurposing approved drugs, high-throughput phenotypic and target-based screening, and optimizing chemical structures of known drugs [58–62]. Utilizing virtual screening, molecular docking, and MD simulation, the study aimed to repurpose drugs targeting the EtfD, initially assessing the reliability of its predicted structure, and predicting its binding site. This process led to the identification of potential drug candidates, including diacerein, levonadifloxacin, and gatifloxacin.

Diacerein, an anthraquinone derivative drug, demonstrates diverse pharmacological effects, encompassing anti-inflammatory, anticancer, antimicrobial, antidiabetic, chondroprotective, nephroprotective, hepatoprotective, and additional beneficial properties [63, 64]. Research indicates its antimicrobial activity against gram-positive cocci in vitro, with transcriptome analysis revealing its inhibition of bacterial growth by targeting oxidative phosphorylation, substance transport, secondary metabolism, and biosynthesis [65]. Additionally, rhein, a metabolite of diacerein, enhances phagocytosis in macrophages, significantly augmenting TNF-α secretion, independent of lipopolysaccharide presence [66]. Furthermore, novel chemicals, developed by optimizing the anthraquinone scaffold of rhein, exhibit promising activity against Mtb while maintaining low toxicity [67]. Given its antimicrobial activity against gram-

**Table 3. Predicted druglikeness and ADMET analysis of the compounds.**

| Property | Compound | | |
|---|---|---|---|
| Physiochemical properties | Diacerein | Levonadifloxacin | Gatifloxacin |
| Formula | C19H12O8 | C19H21FN2O4 | C19H22FN3O4 |
| Molecular weight | 368.29 g/mol | 360.38 g/mol | 375.39 g/mol |
| Num. heavy atoms | 27 | 26 | 27 |
| Num. arom. heavy atoms | 12 | 10 | 10 |
| Fraction Csp3 | 0.11 | 0.47 | 0.47 |
| Num. rotatable bonds | 5 | 2 | 4 |
| Num. H-bond acceptors | 8 | 5 | 6 |
| Num. H-bond donors | 1 | 2 | 2 |
| Molar refractivity | 89.71 | 99.46 | 106.55 |
| TPSA | 124.04 Å$^2$ | 82.77 Å$^2$ | 83.80 Å$^2$ |
| **Lipophilicity** | | | |
| Consensus log $P_{o/w}$ | 1.99 | 2.03 | 1.28 |
| Water solubility | Soluble | Soluble | Very soluble |
| **Pharmacokinetics** | | | |
| GI absorption | High | High | High |
| BBB permeant | No | No | No |
| P-gp substrate | No | Yes | Yes |
| CYP1A2 inhibitor | No | No | No |
| CYP2C19 inhibitor | No | No | No |
| CYP2C9 inhibitor | No | No | No |
| CYP2D6 inhibitor | No | Yes | No |
| CYP3A4 inhibitor | No | No | No |
| Log Kp (skin permeation) | -7.20 cm/s | -7.37 cm/s | -9.12 cm/s |
| **Druglikeness** | | | |
| Lipinski | Yes, 0 violation | Yes, 0 violation | Yes, 0 violation |
| Ghose | Yes | Yes | Yes |
| Veber | Yes | Yes | Yes |
| Egan | Yes | Yes | Yes |
| Muegge | Yes | Yes | Yes |
| Bioavailability score | 0.56 | 0.56 | 0.55 |
| **Medicinal Chemistry** | | | |
| PAINS | 1Alert: quinone_A | 0 Alert | 0 Alert |
| Brenk | 1Alert: phenol_ester | 0 Alert | 0 Alert |
| Synthetic accessibility | 3.08 | 3.75 | 3.47 |
| **Toxicity** | | | |
| Hepatotoxicity | Yes | Yes | Yes |
| Skin sensitization | No | No | No |
| *T.Pyriformis* toxicity (log ug/L) | 0.285 | 0.282 | 0.283 |
| AMES toxicity | No | No | No |
| Minnow toxicity (log mM) | 1.954 | 1.951 | 1.601 |
| Max. tolerated dose (humans) (log mg/kg/day) | 0.636 | 1.224 | 1.158 |
| **Excretion** | | | |
| Total clearance (log ml/min/kg) | 0.348 | 0.549 | 0.693 |
| Renal OCT2 substrate | No | No | No |

**Table 4. Anti-TB activity prediction of top drugs through online server mycoCSM.**

| Compound | Predicted Mtb. MIC (log μM) | Caseum FU (%) | MRTD log (mg/kg/day) |
|---|---|---|---|
| Diacerein | -4.732 | 6.593 | 0.288 |
| Levonadifloxacin | -5.429 | 15.609 | 1.005 |
| Gatifloxacin | -5.488 | 21.732 | 0.929 |
| Rifampicin | -6.312 | 7.493 | 1.106 |
| Isoniazid | -4.942 | 67.2 | 1.166 |

MRTD, Maximum Recommended Therapeutic Dose. Note: Caseum FU (%) represents the predicted ability of compounds to penetrate necrotic tuberculosis foci, with higher values indicating a greater likelihood of penetration into these foci.

positive cocci, inhibition of oxidative phosphorylation, and ability to enhance phagocytosis in macrophages, diacerein shows promise for repurposing against TB. Moreover, the development of new anti-TB compounds through the engineering of rhein adds weight to this potential application.

Levonadifloxacin, a novel broad-spectrum fluoroquinolone, effectively targets challenging infections caused by multidrug-resistant Gram-positive, intracellular, atypical, anaerobic, and Gram-negative bacteria, particularly respiratory pathogens, with superior safety, tolerability, and minimal drug-drug interactions attributed to its lack of CYP interaction [68]. Levonadifloxacin demonstrates superior penetration into alveolar macrophages (AMs) and epithelial lining fluid (ELF), with mean values surpassing those reported for levofloxacin and moxifloxacin. Additionally, its killing effect against susceptible bacteria has been found to be superior to that of comparator quinolones [69]. Levonadifloxacin offers the advantage of effectiveness against resistant organisms while demonstrating a notably low mutation rate [70, 71]. Fluoroquinolones like gatifloxacin, moxifloxacin, and levofloxacin are pivotal in treating drug-resistant TB when combined with standard regimens such as HREZ [72]. Gatifloxacin has achieved an 87% success rate in MDR-TB patients susceptible to the drug, compared to 51% in those with high-level resistance, and has also shown significant impact in treating XDR-TB [73, 74]. Studies indicate that gatifloxacin-based regimens outperform those based on moxifloxacin or levofloxacin [75]. High-dose gatifloxacin-based shorter treatment regimens (STR) are effective for drug-resistant TB, though careful monitoring for hepatotoxicity and QT interval prolongation is essential [76].

In conclusion, this study highlights three potential drugs for repurposing against TB. Notably, gatifloxacin has already been used in anti-tuberculous regimens, while the top two candidates, diacerein and levonadifloxacin, have not yet been utilized as anti-tuberculous drugs. Given the broad-spectrum activity of diacerein and levonadifloxacin, alongside existing evidence for gatifloxacin, further experimental and clinical studies are strongly recommended to evaluate their suitability and clinical efficacy, potentially integrating them into conventional TB treatment regimens. Our future work will prioritize these drugs to assess their anti-TB activity through comprehensive in vitro and in vivo experimentation.

## Supporting information

**S1 Fig. flDPnn and predicted aligned error (PAE).** (a) flDPnn (putative function- and linker-based Disorder Prediction using deep neural network). (b) PAE 2D heatmap.
(PDF)

**S2 Fig. MolProbity Ramachandran plot.**
(PDF)

**S3 Fig. ProSA Z-score.**
(PDF)

**S4 Fig. ProSA residue energies.**
(PDF)

**S5 Fig. Residue interactions with SF4 and surrounding binding site.** (a) Residues interacting with SF4. (b) Interactions of binding site residues with surrounding residues. Residues are depicted as sticks, while iron and sulfur atoms are represented as balls. The color scheme includes carbon (gray), nitrogen (blue), oxygen (red), sulfur of amino acid residues (yellow), sulfur of SF4 (deep red), iron (magenta), hydrogen bonds (cyan), pi sulfur interactions (dark green), and metal acceptor interactions (black).
(PDF)

**S6 Fig. Molecular docking 2D representations.**
(PDF)

**S7 Fig. 2D Interactions of EtfD with diacerein, levonadifloxacin, and gatifloxacin at 0, 50, and 100 ns.** The color scheme follows that of Fig 3d.
(PDF)

**S1 Table. Structure and binding affinity values of top 20 molecules after VS.**
(PDF)

## Author Contributions

**Conceptualization:** Kaleem Arshad.

**Formal analysis:** Kaleem Arshad, Jahanzab Salim.

**Investigation:** Kaleem Arshad, Jahanzab Salim, Muhammad Ali Talat.

**Methodology:** Kaleem Arshad.

**Project administration:** Kaleem Arshad, Nazia Kanwal.

**Resources:** Kaleem Arshad, Jahanzab Salim, Muhammad Ali Talat, Asifa Ashraf.

**Supervision:** Asifa Ashraf, Nazia Kanwal.

**Validation:** Kaleem Arshad, Jahanzab Salim, Muhammad Ali Talat, Asifa Ashraf.

**Visualization:** Kaleem Arshad, Muhammad Ali Talat, Asifa Ashraf.

**Writing – original draft:** Kaleem Arshad.

**Writing – review & editing:** Kaleem Arshad, Jahanzab Salim, Muhammad Ali Talat, Asifa Ashraf, Nazia Kanwal.

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
