## [Decision Letter · Decision Letter 0]

23 Sep 2024

PONE-D-24-33623Integrated virtual screening and MD simulation study to discover potential inhibitors of mycobacterial electron transfer flavoprotein oxidoreductasePLOS ONE

Dear Dr. Arshad,

Thank you for submitting your manuscript to PLOS ONE. After careful consideration, we feel that it has merit but does not fully meet PLOS ONE’s publication criteria as it currently stands. Therefore, we invite you to submit a revised version of the manuscript that addresses the points raised during the review process.

We look forward to receiving your revised manuscript.

Kind regards,

Wagdy M. Eldehna, Ph.d

Academic Editor

PLOS ONE

Journal requirements: 1. When submitting your revision, we need you to address these additional requirements. Please ensure that your manuscript meets PLOS ONE's style requirements, including those for file naming. The PLOS ONE style templates can be found at https://journals.plos.org/plosone/s/file?id=wjVg/PLOSOne_formatting_sample_main_body.pdf and https://journals.plos.org/plosone/s/file?id=ba62/PLOSOne_formatting_sample_title_authors_affiliations.pdf. 2. Please note that PLOS ONE has specific guidelines on code sharing for submissions in which author-generated code underpins the findings in the manuscript. In these cases, all author-generated code must be made available without restrictions upon publication of the work. Please review our guidelines at https://journals.plos.org/plosone/s/materials-and-software-sharing#loc-sharing-code and ensure that your code is shared in a way that follows best practice and facilitates reproducibility and reuse.

Reviewers' comments:

Reviewer's Responses to Questions

**Comments to the Author**

1. Is the manuscript technically sound, and do the data support the conclusions?

Reviewer #1: Yes

Reviewer #2: Yes

2. Has the statistical analysis been performed appropriately and rigorously? 

Reviewer #1: N/A

Reviewer #2: N/A

3. Have the authors made all data underlying the findings in their manuscript fully available?

Reviewer #1: Yes

Reviewer #2: Yes

4. Is the manuscript presented in an intelligible fashion and written in standard English?

Reviewer #1: Yes

Reviewer #2: Yes

5. Review Comments to the Author

Reviewer #1: Dear authors,

The manuscript is well-written and is sound, yet I have few comments and concerns that need to be addressed, which are:

Validating the identified binding site by docking the native substrate for this protein into that site and the other identified binding sites.

MD simulations need to be longer than 50 ns, a minimum of 100 ns is needed to “somehow” ensure reasonable convergence!

It is well-known that the MOA of fluoroquinolones is via inhibiting bacterial DNA gyrase and topo IV, moreover, mutations in these proteins in MtB imparts resistance against those drugs. Accordingly, the presumed anti-MtB activity of levonadifloxacin, and gatifloxacin as being potential inhibitors of the mycobacterial electron transfer flavoprotein oxidoreductase would be questionable!!

Reviewer #2: Firstly, regarding the topic:

It is a good idea in terms of concept and execution.

It is written in strong English.

Secondly, it is possible to enhance the quality of the attached images as their current quality is very poor.

Thirdly, concerning MD simulation:

The presence of major fluctuations during the run indicates instability of the compound within its target site.

Therefore, you should include images of the compounds within the target at the beginning, middle, and end of the run for each one.

6. PLOS authors have the option to publish the peer review history of their article (what does this mean?). If published, this will include your full peer review and any attached files.

Reviewer #1: **Yes: **Nizar Ali Al-Shar'i

Reviewer #2: **Yes: **Abdulrahman M. Saleh

---

## [Author Response · Author response to Decision Letter 0]

5 Oct 2024

Wagdy M. Eldehna

Academic Editor

PLOS ONE

Dear Eldehna,

We would like to express our sincere gratitude for the opportunity to submit our manuscript titled "Integrated virtual screening and MD simulation study to discover potential inhibitors of mycobacterial electron transfer flavoprotein oxidoreductase" to PLOS ONE. We deeply appreciate the insightful comments and suggestions provided by you and the reviewers, which have greatly contributed to improving our work.

We have carefully considered each comment and made the necessary revisions to the manuscript. Below is our point-by-point response to the reviewers' comments:

Reviewer #1:

Comment 1: Recommendation to validate identified binding sites by docking the native substrate.

The COACH meta server predicted the iron-sulfur cluster (SF4) as the ligand for the highest-scoring predicted binding site. This prediction was further supported by structure similarity clustering in the AlphaFold Protein Structure Database, using MMseq2 and Foldseek, which identified a cluster related to EtfD. This cluster comprised proteins with iron-sulfur binding domains and oxidoreductase activity across diverse bacterial species, some containing cysteine-rich domains. This finding aligned with the established role of EtfD in energy production and its [4Fe-4S] ferredoxin-type iron-sulfur binding domain, as confirmed by InterPro analysis. Additionally, a BLASTp search of the PDB yielded no closely related experimentally determined structures, with only three (5ODC_B, 5ODC_C, 7BKB_C) showing low query coverage and identity, yet all are heterodisulfide reductases containing SF4 as their native substrate. These findings, along with COACH predictions and the recognized function of EtfD as an electron transfer flavoprotein-oxidoreductase, strongly support our conclusion that SF4 is the native ligand for the predicted binding site.

We validated the predicted binding site by docking the native substrate SF4 using AutoDock4. The docking grid (spacing 0.375 Å, grid size 22.5 Å x 22.5 Å x 22.5 Å) covered all binding sites predicted by the COACH server, with specific coordinates (X = 4.2938, Y = 5.8156, Z = -6.6638). We performed 250 genetic algorithm runs and 25 million energy evaluations ('long' option), determining a binding affinity of -4.47 kcal/mol for SF4. This confirms the predicted binding site as the valid site for SF4.

In summary, we first established confidence that SF4 is the native substrate of the predicted binding site, which we subsequently confirmed through rigorous molecular docking.

Comment 2: MD simulations require a minimum duration of 100 ns for reasonable convergence.

We appreciate your insightful comment regarding the duration of the molecular dynamics (MD) simulations. In response, we performed MD simulations for a total of 100 ns, restarting the simulations from the 50 ns mark. We are pleased to report that this extension allowed our top three drug candidates to converge to a reasonable extent, yielding promising results. Thank you for highlighting this important aspect of our study.

Comment 3: Concerns regarding the mechanism of action of fluoroquinolones and potential resistance in mycobacterium tuberculosis.

Thank you for highlighting this important aspect. We acknowledge that the well-known mechanism of action (MOA) of fluoroquinolones (FQNs) involves inhibiting DNA gyrase and topoisomerase IV. However, several studies have also demonstrated a correlation between FQNs lethality and the formation of reactive oxygen species (ROS) (Dwyer et al. 2007; Hong et al. 2019; 2020). This additional mechanism suggests that FQNs may exert bactericidal effects beyond targeting DNA replication machinery, potentially impacting other cellular pathways. For instance, FQNs form complexes with Cu/Zn-superoxide dismutase (SOD) through hydrogen bonds and van der Waals forces, leading to structural and functional changes in the enzyme that induce oxidative stress, suggesting that quinolones may target proteins beyond their primary mechanism of action (Pan et al. 2016).

Several studies have shown that FQN inhibition of gyrase triggers the upregulation of DNA damage response and repair genes, as well as increased expression of genes related to superoxide stress and iron-sulfur cluster synthesis. Additionally, fluoroquinolone administration generates hydroxyl radicals, indicating that these drugs induce oxidative stress, contributing to bacterial cell death (Dwyer et al. 2007).

One possible mechanism by which oxidative stress can kill bacteria involves the vulnerability of cytosolic proteins containing iron-sulfur (Fe–S) clusters to oxidative damage (Imlay 2003; 2006). The ROS-mediated decomposition of Fe–S clusters releases ferrous (Fe²⁺) iron into the cytoplasm, where elevated concentrations can cause DNA damage directly or by generating additional oxidative molecules (Rai et al. 2001; Touati 2000). Hydroxyl radicals, highly destructive reactive oxygen species (ROS), are produced through the Fenton reaction, in which ferrous iron facilitates the reduction of hydrogen peroxide, potentially leading to bacterial demise.

Based on these observations, it is plausible that levonadifloxacin and gatifloxacin can target electron transfer flavoprotein oxidoreductase (EtfD), leading to the accumulation of reduced flavin adenine dinucleotide (FAD) that induces reductive stress, impairing metabolism, causing protein aggregation, generating ROS, and ultimately resulting in mycobacterial demise (Mavi, Singh, and Kumar 2020).

Thus, we believe the potential of levonadifloxacin and gatifloxacin as inhibitors of mycobacterial electron transfer flavoprotein oxidoreductase (EtfD) remains plausible and warrants further exploration.

Reviewer #2:

Comment 1: Suggestion to enhance image quality.

Thank you for your positive feedback regarding the concept and execution of our study. We appreciate your suggestion to enhance the quality of the images. In response, we have increased the resolution of all images to 600 dpi. We apologize for the previous images; while we had individual images of high quality (600 to 1200 dpi), we encountered difficulties merging them, which resulted in poor quality. We have since utilized GIMP software to merge the figures, and they are now of publication quality. Thank you for highlighting this issue.

Comment 2: Request for images illustrating compound stability at the target site throughout MD simulations.

We are pleased to inform you that, in line with the suggestion from another reviewer, we extended the duration of the molecular dynamics (MD) simulations to 100 ns. This extension resulted in the convergence of our compounds within the binding site, with no major fluctuations observed post-convergence. Additionally, as you recommended, we have included 2D representations illustrating the interactions between the drug candidates and the target protein, EtfD, at the beginning (0 ns), middle (50 ns), and end (100 ns) of the simulations (S7 Fig). Thank you for your valuable feedback.

Final words:

We sincerely appreciate the time and effort you and the reviewers have dedicated to evaluating our manuscript. Your constructive feedback has significantly improved the quality of our work. We believe that the revisions and additional data have addressed the reviewers' concerns comprehensively.

We are excited about the potential implications of our findings and look forward to the possibility of our work contributing to the field. Thank you for considering our revised manuscript for publication in PLOS ONE. We are hopeful that it will meet your expectations and the journal's standards.

Kind regards,

Dr. Kaleem Arshad

Superior University, Lahore, Pakistan

Kaleemarshad630@gmail.com

References:

Dwyer, Daniel J., Michael A. Kohanski, Boris Hayete, and James J. Collins. 2007. “Gyrase Inhibitors Induce an Oxidative Damage Cellular Death Pathway in Escherichia Coli.” Molecular Systems Biology 3 (1): 91.

Hong, Yuzhi, Qiming Li, Qiong Gao, Jianping Xie, Haihui Huang, Karl Drlica, and Xilin Zhao. 2020. “Reactive Oxygen Species Play a Dominant Role in All Pathways of Rapid Quinolone-Mediated Killing.” Journal of Antimicrobial Chemotherapy 75 (3): 576–85.

Hong, Yuzhi, Jie Zeng, Xiuhong Wang, Karl Drlica, and Xilin Zhao. 2019. “Post-Stress Bacterial Cell Death Mediated by Reactive Oxygen Species.” Proceedings of the National Academy of Sciences 116 (20): 10064–71.

Imlay, James A. 2003. “Pathways of Oxidative Damage.” Annual Reviews in Microbiology 57 (1): 395–418.

———. 2006. “Iron‐sulphur Clusters and the Problem with Oxygen.” Molecular Microbiology 59 (4): 1073–82.

Mavi, Parminder Singh, Shweta Singh, and Ashwani Kumar. 2020. “Reductive Stress: New Insights in Physiology and Drug Tolerance of Mycobacterium.” Antioxidants & Redox Signaling 32 (18): 1348–66.

Pan, Xingren, Pengfei Qin, Rutao Liu, Jianfeng Li, and Fucui Zhang. 2016. “Molecular Mechanism on Two Fluoroquinolones Inducing Oxidative Stress: Evidence from Copper/Zinc Superoxide Dismutase.” RSC Advances 6 (94): 91141–49.

Rai, Priyamvada, Timothy D. Cole, David E. Wemmer, and Stuart Linn. 2001. “Localization of Fe2+ at an RTGR Sequence within a DNA Duplex Explains Preferential Cleavage by Fe2+ and H2O2.” Journal of Molecular Biology 312 (5): 1089–1101.

Touati, Danièle. 2000. “Iron and Oxidative Stress in Bacteria.” Archives of Biochemistry and Biophysics 373 (1): 1–6.

---

## [Decision Letter · Decision Letter 1]

15 Oct 2024

Integrated virtual screening and MD simulation study to discover potential inhibitors of mycobacterial electron transfer flavoprotein oxidoreductase

PONE-D-24-33623R1

Dear Dr. Arshad,

We’re pleased to inform you that your manuscript has been judged scientifically suitable for publication and will be formally accepted for publication once it meets all outstanding technical requirements.

Kind regards,

Wagdy M. Eldehna, Ph.d

Academic Editor

PLOS ONE

Additional Editor Comments (optional):

Reviewers' comments:

Reviewer's Responses to Questions

**Comments to the Author**

1. If the authors have adequately addressed your comments raised in a previous round of review and you feel that this manuscript is now acceptable for publication, you may indicate that here to bypass the “Comments to the Author” section, enter your conflict of interest statement in the “Confidential to Editor” section, and submit your "Accept" recommendation.

Reviewer #1: All comments have been addressed

Reviewer #2: All comments have been addressed

2. Is the manuscript technically sound, and do the data support the conclusions?

Reviewer #1: Yes

Reviewer #2: Yes

3. Has the statistical analysis been performed appropriately and rigorously? 

Reviewer #1: N/A

Reviewer #2: N/A

4. Have the authors made all data underlying the findings in their manuscript fully available?

Reviewer #1: Yes

Reviewer #2: Yes

5. Is the manuscript presented in an intelligible fashion and written in standard English?

Reviewer #1: Yes

Reviewer #2: Yes

6. Review Comments to the Author

Reviewer #1: Thank you for addressing my comments.

A final comment: Please check Fig. 5a (the RMSD plot) the behavior of the simulated complexes at the extra 50 ns is different from the first 50 ns, make sure you are fitting the frames to the same reference structure, or if there was a restraining force being applied to the system.

Reviewer #2: Dear Author,

You have addressed the proposed revisions in an organized and effective manner. The changes you made have significantly improved the clarity and quality of the manuscript.

7. PLOS authors have the option to publish the peer review history of their article (what does this mean?). If published, this will include your full peer review and any attached files.

Reviewer #1: No

Reviewer #2: **Yes: **Abdulrahman M. Saleh

---

## [Editor Report · Acceptance letter]

6 Nov 2024

PONE-D-24-33623R1 

PLOS ONE

Dear Dr. Arshad, 

I'm pleased to inform you that your manuscript has been deemed suitable for publication in PLOS ONE. Congratulations! Your manuscript is now being handed over to our production team.

Kind regards, 

on behalf of

Dr. Wagdy M. Eldehna 

Academic Editor

PLOS ONE